# communications
# earth & environment

# Routes to reduction of phosphate by high-energy events

Luca Bindi [1], Tian Feng [2] & Matthew A. Pasek [2✉]

Phosphate minerals such as those in the apatite group tend to be the dominant forms of phosphorus in minerals on the Earth's surface. Phosphate can be reduced to phosphides during high-energy events, such as lightning and impacts. Here we show that, in addition to formation of metal phosphides, a new compound was formed by lightning in a fulgurite from New Port Richey, Florida, USA. A calcium phosphite material, ideally $CaHPO_3$, was found in spherules mainly consisting of iron silicides that formed by lightning-induced fusion of sand around a tree root. This phosphite material bears a phosphorus oxidation state intermediate of that of phosphides and phosphates in a geologic sample and implicates phosphites as being potentially relevant to other high-energy events where phosphorus may partially change its redox state, and material similar to this phosphite may also be the source of phosphite that makes up part of the phosphorus biogeochemical cycle.

[1] Dipartimento di Scienze della Terra, Università di Firenze, Via La Pira 4, I-50121 Firenze, Italy. [2] Department of Geosciences, University of South Florida, Tampa, FL 33620, USA. ✉email: mpasek@usf.edu

The mineralogy of the element phosphorus (P) is generally perceived as being limited to phosphates under terrestrial conditions. Cosmically, P-mineralogy includes phosphide minerals in which phosphorus mainly forms binary compounds with a metal such as Fe or Ni. The mineral schreibersite, $(Fe,Ni)_3P$, is an example of such a binary solid and is cosmically the most abundant phosphide. Terrestrial occurrences of schreibersite and other phosphides have since been discovered within pyrometamorphic rocks[1,2], and especially within fulgurites[3,4], which are glasses formed by cloud-to-ground lightning. The presence of phosphides within fulgurites parallels the existence of phosphides within impact rocks such as tektites[5], with impact cratering also being the likely route to forming phosphides in lunar rocks[6], and in impact melts on other meteorites[7,8].

Phosphorus has thus far been found in mineral form in only two oxidation states: as +5 in phosphates, and as ~−1 in phosphides. This latter oxidation state is based on the binding energy of inner electrons of the P atom in phosphides[9], which is confirmed by ab initio calculations[10]. Notably absent from known P-mineralogy are any other oxidation states between these, wherein the P anion is a reduced phosphorus oxyacid. Salts with P in a +1 oxidation state (hypophosphite), +3 oxidation state (phosphite), and +4 oxidation state (hypophosphate) have been synthesized but are not known from naturally-occurring geologic samples. Despite the apparent lack of reduced phosphorus oxyacids in mineral form, reduced phosphorus compounds are not rare in the geo-environment. For example, the gas phosphine ($PH_3$) has also been recognized to be a trace gas in many environments[11–13], and has even been implicated as a biosignature gas on Venus[14]. More specifically to the case of potentially mineral-forming P-compounds, between 1 and 10% of all microbially lineages can use phosphite (and often hypophosphite) as sole P sources[15], demonstrating these ions have sources in the environment[16]. In addition, reduced phosphorus oxyacids have been found in geothermal spring water and anaerobic water in Florida where their origin is likely microbial[17,18]. Within rock samples phosphite has been found in extracts of fulgurites[19], in serpentinites[20], and in extracts of Archean rock samples[21,22], though the mineral source of such a phase is unclear. Therefore, reduced P oxyacids must exist in solid form of some sort in the environment, either as a pure compound such as $CaHPO_3$, or with the phosphite exchanging for other ions such as phosphate or sulfate. The failure to identify a specific mineral form of reduced phosphorus oxyacid in the environment is problematic given the widespread evidence for phosphorus redox in biogeochemistry[16], but may be due instead to an incomplete search for such material.

Due to their rapid formation which results in lithologies that may vary significantly with respect to temperature, pressure, and $f_{O_2}$ over a few millimeters of distance, fulgurites are an ideal material in which to search for reduced P-compounds. Phosphides such as schreibersite have been identified in multiple fulgurites: from Michigan[3], Pennsylvania[19], and Illinois[4]. In addition, even more reduced material such as iron silicides are recognized in multiple fulgurites[3,23–29]. Such material is presumed to form from the intensely reducing environment possibly generated by a combustion reaction of in situ organics[3,30]. Pasek and Block[19] also reported phosphite in extracts of fulgurites, which were attributed in one fulgurite to phosphides, which react with water to release phosphite as a metastable solute[31], but in others the source of phosphite could not be wholly identified.

Here we identify (1) a novel form of crystalline P within a fulgurite from New Port Richey, Florida, USA; and (2) propose that the reduction of phosphate to more reduced forms takes place through at least two separate pathways and that both may be relevant to the planetary distribution of phosphorus[32,33]. The

New Port Richey (hereafter NPR) fulgurite was obtained from private sellers who found the fulgurite on their property in a residential neighborhood in New Port Richey, Florida (28.248 N, 82.718 W) the summer of 2012 after a thunderstorm. We then purchased the fulgurite in late 2012. The fulgurite formed in a partially drained sandy soil, composed primarily of quartz sand and clay[34], and was about 500 g in total mass. In addition, as the area receives heavy rain, roots are often coated with an iron oxide plaque[35–37], which is composed of a 1–2-mm-thick rind of iron oxide that cements quartz sand particles (Supplementary Fig. 1).

The fulgurite is a type II fulgurite[25] with a massive, glassy wall that surrounds the central void (Fig. 1a). The fulgurite's color ranges from white to dark gray. The most unusual feature of the fulgurite is the presence of large (~1 cm diameter) spherules (Fig. 1b) of a material with a dull gray metallic luster that are embedded in the glassy matrix of the fulgurite. These spherules have a density of 4.9–5 $g/cm^3$ and are not attracted to a magnet.

## Results

The New Port Richey fulgurite bears both silicate and reduced (silicide) compositional regions. The silicate fraction of the New Port Richey fulgurite is volumetrically more common and is composed of glass with two major compositions (Fig. 1c): one exclusively $SiO_2$, likely relict quartz sand grains, and the other composed mostly of Ca, Al, and Si oxides (Supplementary Table 1). Within the silicate lithologies are small droplets (<15 μm) composed of P-rich iron metal (Fig. 1d). Such droplets have been reported previously in other fulgurites[3,4,19].

The large metal spherules within the NPR fulgurite consist of silicides, specifically with the composition of FeSi and $FeSi_2$, in addition to a material that is darker in contrast in back-scattered electron (BSE) imaging that bears Ca and P and is amorphous to X-ray diffraction (Fig. 1e, f). The silicides also bear 0.45–1.25 wt.% P. The groundmass of the spherules is composed of $FeSi_2$ (84% by volume), in which euhedral crystals of FeSi (11%) grow, often together with the CaP material (5%). The nominal cationic composition of the spherule is thus $Si_{63}Fe_{33}Ca_2P_2$. The CaP material bears vermiform FeSi with a width of ~1 μm. The CaP material is most closely associated with the FeSi material, which grows as euhedral crystals (30–70 μm) that are larger than the average FeSi crystals in the $FeSi_2$ groundmass, or more compact (with less extreme maximum compared to minimum dimensions). Though the CaP material is in contact with the $FeSi_2$ groundmass, crystal growth of $FeSi_2$ within the CaP material is less common. Inside the CaP material there are very rare (less than 20 μm sized) crystal fragments that turned out to be a new crystalline phosphite (see the dashed circle in Supplementary Fig. 2).

Given the similarity in chemical composition and unit-cell values, the structure of the new phosphite was refined starting from the atom coordinates given by Phillips and Harrison[38] for the $P4_32_12$ structural model of synthetic $CaHPO_3$. Phillips and Harrison[38] describe the three-dimensional structure of synthetic $CaHPO_3$ as having $CaO_7$ capped distorted trigonal prisms, with pseudopyramids of $HPO_3$ sharing corners and edges. The formula of the phosphite phase in the NPR fulgurite is $(Ca_{0.91}Fe_{0.09})HPO_3$. Although there is a thin boundary between anthropogenic and natural fulgurites (see below), here we first report a phosphite compound which represents the reduced version of monetite ($CaPO_3OH$), as the phosphite ($HPO_3^{2-}$) anion bears a +3 oxidation state for P. To the best of our knowledge, $CaHPO_3$ is also the first fulgurite-occurring crystalline material wherein P is not a phosphate or a phosphide.

We attempted to replicate the $CaHPO_3$ mineral formation by adding $CaHPO_4$ to a mixture of Fe and Si (0.1:1:2 molar ratio)

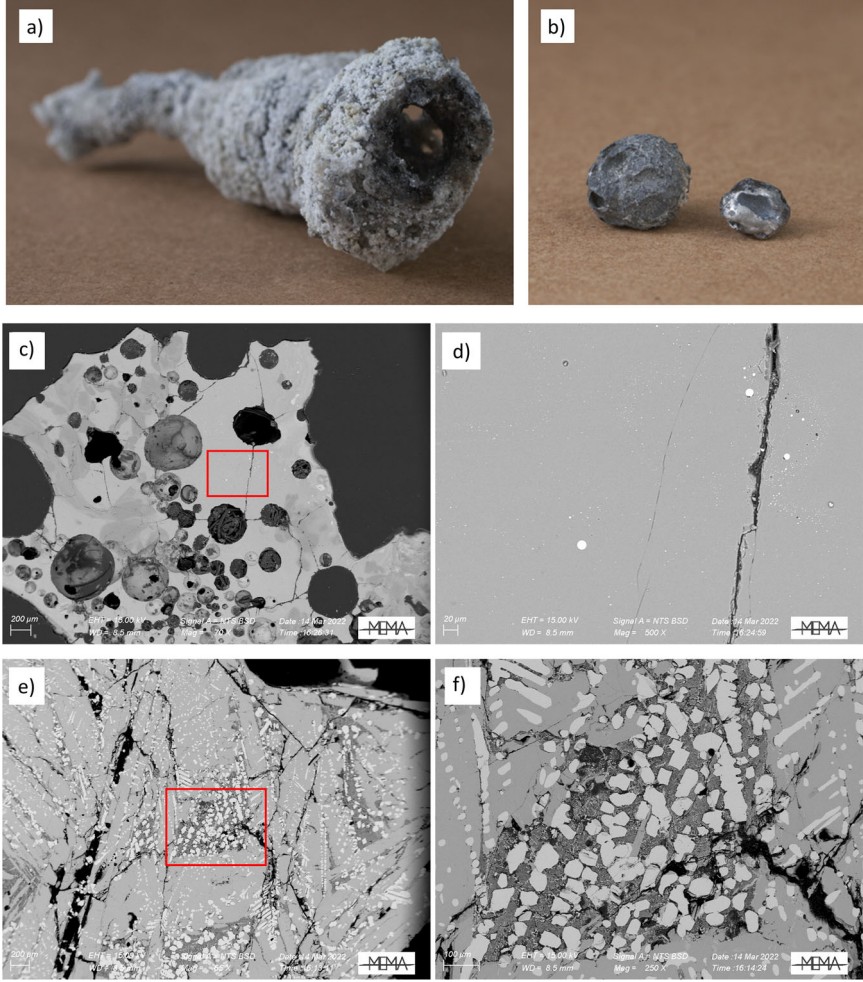

**Fig. 1 The New Port Richey fulgurite images and microscopy. a** Glassy tubes that consist of a glassy melt surrounding an internal void, in turn surrounded by a crust consisting of cemented sand grains. The diameter of the fulgurite is 2 cm, and length is 7 cm. **b** Spherules of gray, metallic material within the fulgurite with diameters of 1.1 cm (left) and 0.5 cm (right). **c** BSE image of the glass of the NPR fulgurite. Varied lithologies of the glass include a darker material (left) composed of $SiO_2$, and a lighter material (Ca, Al-rich silicate). Within this glass (red rectangle is the region expanded) are **d** spherules of iron metal enriched in phosphorus. **e** BSE image of the large metallic spherules of the NPR fulgurite. These consist of $FeSi_2$ (medium gray), FeSi (light gray), and a Ca–P–O material that includes $CaHPO_3$ (dark gray). **f** The Ca–P–O material is mostly in contact with FeSi.

and heating to 1000 °C for 8 h. The phosphate was demonstrated to form reduced P-compounds upon extraction with $Na_4EDTA$ solution by NMR (nuclear magnetic resonance) spectroscopy (Fig. 2) that demonstrates the formation of $P^{3+}$ and $P^{4+}$ species. However, such chemistry naturally arises from the extraction of P-compounds from $Fe_3P$ (or schreibersite), and the ratio of these species (1:5:5:1 of hypophosphate: phosphate: phosphite: pyrophosphate, see ref. [39]) implicates that—rather than reducing $CaHPO_4$ to $CaHPO_3$—reduction of $CaHPO_4$ to $Fe_3P$ occurred instead. If we assume that the reaction velocity doubles with every 30 °C increase in temperature (corresponding to an activation energy of 20 kJ/mol), then at 1220 °C—where $FeSi_2$ would have melted—we would expect to see the reduction of $CaHPO_4$ to $Fe_3P$ over the course of 5 min. Given that this did not occur in the NPR fulgurite, that implies the reduction timescale was less than 5 min.

The NPR fulgurite bears a crystalline $CaHPO_3$ material that may be the first mineral bearing a phosphite ($HPO_3^{2-}$, $P^{3+}$) group to the best of our knowledge. Notably, this fulgurite is the second to have formed from New Port Richey, FL, USA (contrasted with that reported in ref. [40]). However, fulgurites can occur in material that is also influenced by either (1) artificial target material such as pavement or conductor metal, or

(2) formed through downed powerlines that introduce energy to the environment slowly and over time[40]. Notably, the NPR fulgurite shows no indication of the former as there is no aluminum or copper in a metallic material, and the groundmass of the fulgurite is consistent with local soil compositions[34]. The source of the discharge to be an artificial source may be unlikely due to the incomplete reduction of phosphate to phosphide within the $FeSi/FeSi_2$ droplets, as reduction should be complete if the system attained equilibrium. The reduction experiments described above show $CaHPO_4$ to $Fe_3P$ would likely have occurred as temperatures exceeded 1220 °C (the melting point of $FeSi_2$) under relatively short timescales (~5 min) for this fulgurite, presuming a reaction rate doubling every increase of temperature of 30 °C. That the reduction of $CaHPO_4$ to $Fe_3P$ did not occur implicates a shorter (<100 s) heating timescale, consistent with a lightning strike.

A metallic silicide spherule belonging to the NPR fulgurite was studied by high-resolution multiscale nanotomography. Fig. 3 reports the color-coded 3D thickness distribution of the pore network inside the cement sample (see Supplementary Video 1) together with the thickness distribution plot of the analysis (inset in Fig. 3). The porosity also a directionality to it, which may be

the same as the orientation of the flow of gas out of the fulgurite. Figure 4 shows a virtual cut of the spherule evidencing melting and recrystallization structures.

The reported presence of much larger iron silicides[41,42] in geologic specimens suggests that large spherules are formed by lightning and may be the source of these silicides that have been reported in various locations. For these reasons, we argue the CaHPO₃ could be a natural material and therefore represent a member of a new mineral group, one predicted to occur in the environment from biology and biochemistry[15].

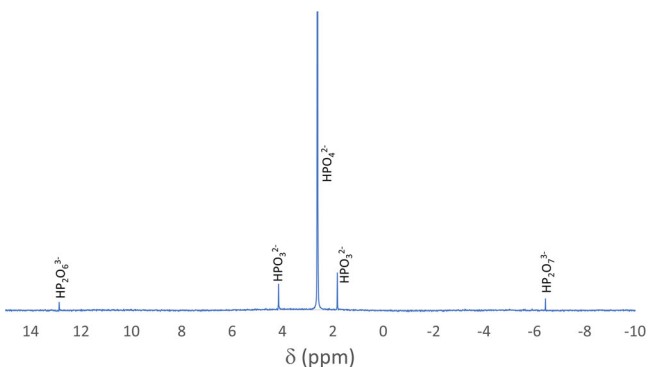

**Fig. 2 ³¹P NMR of Fe–Si–CaHPO₄ (1:2:0.1 by mole) mixture heated to 1000 °C for 8 h.** The NMR spectrum, in which individual anions are referenced to a frequency of 161.907 MHz for H₃PO₄ (0 ppm), identifies P species based on parts per million deviations (δ) from this frequency. The peaks correspond to (left to right) hypophosphate (P⁴⁺), phosphite (P³⁺), phosphate (P⁵⁺), and pyrophosphate (P⁵⁺). Note that phosphite is a broad doublet with a J-coupling constant of 570 Hz when coupled to ¹H. About 7% of the phosphorus in this NMR spectrum has been changed from HPO₄²⁻, which is consistent with reduction of phosphate to phosphide. Phosphide, specifically Fe₃P, reacts with water to form a 1:5:5:1 mixture of hypophosphate: phosphate: phosphite: pyrophosphate. In contrast, a change to CaHPO₃ would yield only phosphite. This suggests that reduction of phosphate to phosphite occurs much faster than the timescale of 8 h reaction, proceeding to Fe₃P within 8 h.

## Discussion

The NPR fulgurite presents an unusual mineralogical paradox: the more reduced phosphorus mineral (P¹⁻ as a metal phosphide) is in contact with a silicate glass, but the more oxidized phosphorus material—the calcium phosphite (P³⁺)—is in direct contact with extremely reduced iron silicides. Such a difference is especially exemplified by the $f_{O_2}$ mineral buffer diagram (Fig. 5) that demonstrates that the formation of CaHPO₃ should not occur in contact with FeSi or FeSi₂. We examine the cause of this disparity in P chemical oxidation state and the implications of its formation.

The presence of silicides within fulgurites has been reported previously[29]. The size of the crystals of the silicides is comparable to the Winans Lake fulgurite[3], wherein crystals of silicides ~10–100 μm wide were found within metal spherules. Compositionally, these silicides also match those reported in the Houghton Lake fulgurite[28], in which FeSi and FeSi₂ formed from a eutectic melt within spherules. The size of the silicide-bearing spherules is much larger than those found previously[25] by about a factor of 10×, which were previously the largest spherules reported in the literature from fulgurites. The size of these spherules (1 cm), however, is comparable to those reported as purported distal impact ejecta[41].

The formation of the iron silicide spherules in the NPR fulgurite likely occurred during the combustion of roots coated with iron oxide plaque (see Supplementary Fig. 3 for details on the reduction of Fe₂O₃ and SiO₂ by organics). Iron oxide plaques form around roots, and cement sand as lithotrophic bacteria oxidize Fe(II) in wetlands environments[35], and such plaques are known to concentrate phosphate[43], including as calcium phosphates such as brushite[44]. During the lightning strike, the combustion of organic root matter likely drove extreme reduction of the iron oxide plaque, melting it as FeSi₂ formed (T > 1220 °C), still well below the melting point of calcium phosphate (>1600 °C[45]). Calcium phosphate, entrained within the iron silicide, was subjected to the extreme reduction, and locally formed the CaHPO₃ material. Given that the CaHPO₃ is in contact with FeSi, and FeSi is formed under somewhat more oxidizing conditions than FeSi₂ (Fig. 5), perhaps the reduction of CaHPO₄ to

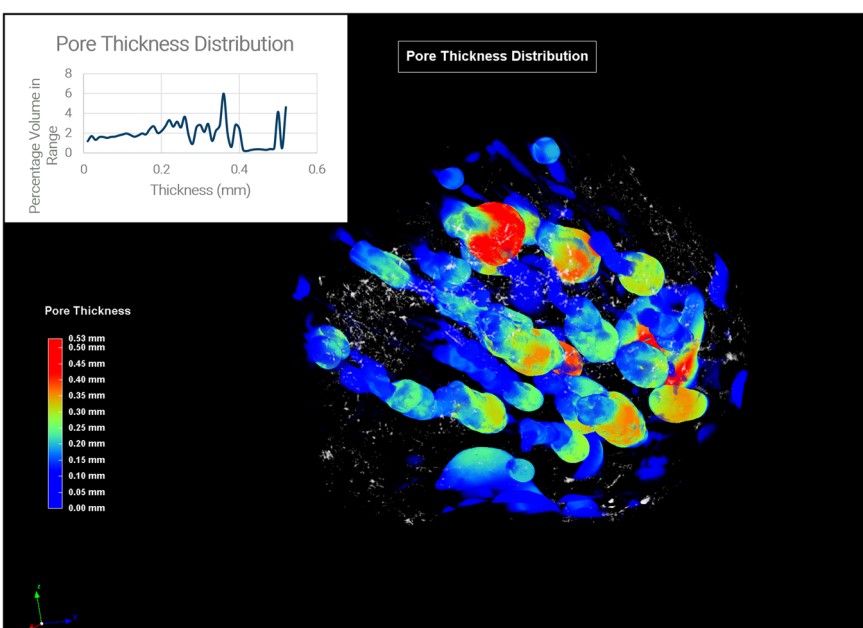

**Fig. 3 Nanotomography of the iron silicide spherule.** The scan in one field of view allows to visualize the color-coded 3D thickness distribution of the pore network inside the sample. The thickness distribution plot of the analysis is shown in the inset.

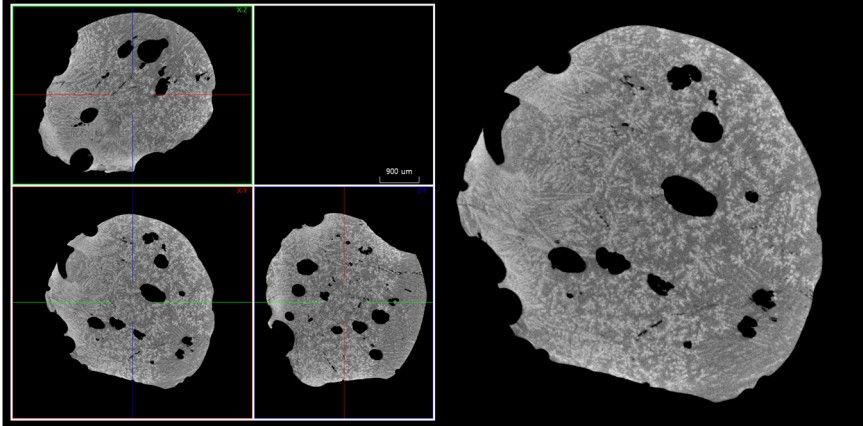

**Fig. 4 Virtual cut of the iron silicide spherule.** The spherule shown in Fig. 3 has evidence of melting and recrystallization structures.

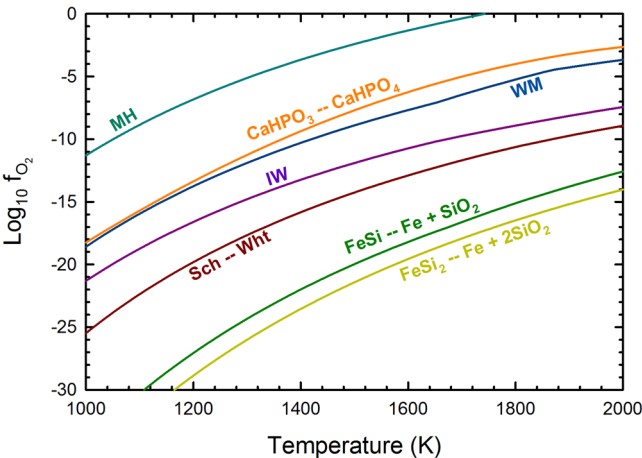

**Fig. 5 Oxygen fugacity diagram.** Oxygen fugacity is buffered by minerals (MH magnetite/hematite, WM wustite/magnetite, IW iron/wustite) that demonstrates the reduction of calcium phosphate to calcium phosphite ($CaHPO_3$ – $CaHPO_4$) should occur at much higher $f_{O_2}$ than other reduction reactions, including the transformation of schreibersite (Sch, nominally $Fe_3P$) to whitlockite (wht, nominally $Ca_3(PO_4)_2$). The transformation of whitlockite to schreibersite should also occur readily if silicates are reduced to silicides.

$CaHPO_3$ was accompanied by the oxidation of $FeSi_2$ to FeSi and $SiO_2$. In addition, as $Fe^{2+}$ is ~9% of the divalent cation of the $CaHPO_3$ material ($Ca_{0.91}Fe_{0.09}HPO_3$) iron as FeSi may have served as a reductant as well. Elemental mapping of the $CaHPO_3$ material (Fig. 6) demonstrates that both Si and O are present within this substance. Experimentally heating $CaHPO_4$ with Fe and Si at 1000 °C did not produce $CaHPO_3$ but instead produced $Fe_3P$ after 8 h, suggesting that rapid heating and cooling of this material—such as what occurs in a lightning strike on the timescale of 60 seconds or less—is necessary to make $CaHPO_3$.

The NPR fulgurite bears two distinct, reduced P phases- the $CaHPO_3$ in the $FeSi/FeSi_2$ spherule, and P dissolved within Fe as much smaller droplets within the silicate glass. These likely represent two separate reduction pathways for phosphorus. The P-rich iron spherules in silicatic glass (Fig. 1c, d) are smaller (15 μm) and their spherical shape and higher P concentration along the edges of the spheres[46] indicate that molten iron reacted with P at high temperature ($T > 1538$ °C), probably as volatilized P, through phosphidation of the metal. The spherical shape (Fig. 1d) also indicates that iron melted. Metal phosphidation has been proposed to be a kinetically favored route to phosphate

reduction that may be the source of metal phosphides on the Moon[47]. This route involves volatilization of P from phosphates to form P and $P_2$ gases under reducing (but not extremely reducing) conditions, followed by reaction of P gas with metal to form metal phosphides, a process that occurs rapidly[48].

In contrast, the $CaHPO_3$ mineral likely formed from the reduction of calcium phosphate, possibly as either amorphous calcium phosphate, or brushite ($CaHPO_4·2H_2O$). Brushite can dehydrate to monetite ($CaHPO_4$) with heating and does so rapidly (timescale of <1 min for similar Mg-phosphates[49]) at temperatures >200 °C[49,50]. Brushite reduction and dehydration, as:

$$CaHPO_4 \cdot 2H_2O = CaHPO_3 + 2H_2O + 1/2\,O_2$$

was approximated using limited thermodynamic data[19] but should occur under more oxidizing conditions than at the wustite-magnetite $f_{O_2}$ buffer. In contrast, the FeSi or $FeSi_2$ oxidation buffer[3] is well below the iron–wustite buffer, just slightly above the Si–$SiO_2$ buffer. This suggests that the reduction of phosphate to phosphide proceeds through a phosphite intermediate, which did not complete due to the relatively short duration of heating (<1 min) during the lightning strike fusion of the soil. The $CaHPO_3$ compound is thus out of equilibrium with its surrounding assemblage and should have been reduced to phosphide.

To this end, the reduction of phosphate proceeds via two routes (Fig. 7): a high temperature (>1700 °C) reduction wherein Fe metal forms and is rapidly phosphidized by P gas at moderately low $f_{O_2}$, and a lower temperature (>1220 °C) reduction wherein P as calcium phosphates is entrained within extremely reduced material and is rapidly reduced to phosphites but kinetically inhibited from forming the ultimate reduced materials, the phosphides.

Although the production of $CaHPO_3$ from $CaHPO_4$ in contact with FeSi likely has few direct counterparts in natural geologic samples (as silicides are rare), the identification of a likely natural, crystalline form of a $P^{3+}$ material is important to understanding the role of redox in the phosphorus cycle. Hess et al.[4] specifically call out the lightning reduction of phosphate to have been a widespread phenomenon on the early Earth potentially transforming thousands of kg of phosphate to phosphides and phosphites annually, which would have rivaled meteoritic sources. These materials may have been prebiotically important[31,51]. Pasek and Block[19] argue lightning would have been a less important (~3000 kg/yr) source of reduced P on the modern Earth, but the relatively facile reduction of phosphate to phosphite (Fig. 5) indicates that phosphites may be present in other geologic or cosmic samples subjected to reduction but that

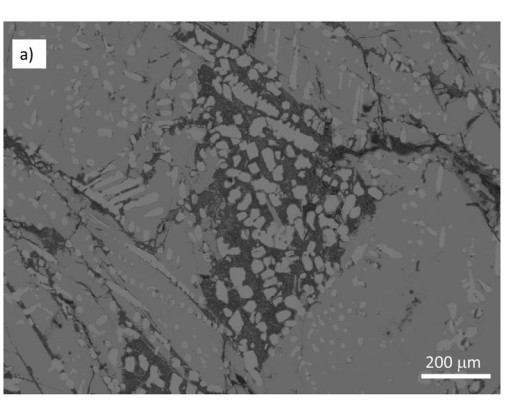

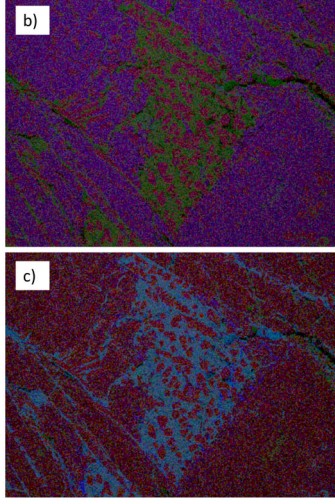

**Fig. 6 Elemental maps of the Ca–P–O material.** The Ca–P–O material (BSE image in **a**) is shown in three-element RGB maps, with (**b**) Fe as red, O as green, and Si as blue, and (**c**) Fe as red, O as green, and P as blue. The Ca–P–O material is associated with some Si and O, suggesting its formation may be coupled to FeSi or FeSi$_2$ oxidation.

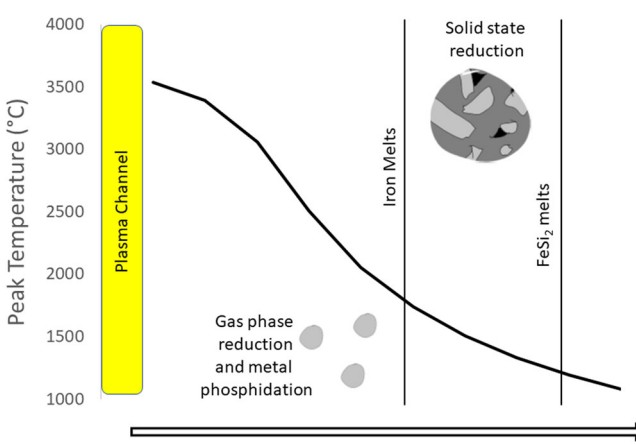

**Fig. 7 Schematic of the two routes to phosphate reduction within the NPR fulgurite.** The high-temperature reduction is rapid due to gas-solid reactions, and occurs closer to the plasma discharge, and the low-temperature reduction occurs in the solid phase and is likely kinetically inhibited from proceeding to form phosphides. Known temperature points are the melting point of the Ca–Al–Si matrix (1195 °C), the melting point of FeSi$_2$ (1220 °C), the melting point of iron metal (1538 °C), the melting point calcium phosphate (>1600 °C), and the melting point of SiO$_2$ (1710 °C). SiO$_2$ is glass near the interior of the fulgurite, whereas it is a crystalline material on the exterior, showing a radial temperature profile that has been reported previously[25].

do not reach very high temperatures. For instance, phosphates within pallasites or iron meteorites may equilibrate with iron metal to transform into phosphite minerals. Phosphite is also found within serpentinites[20], and phosphites such as CaHPO$_3$ (specifically the Mg/Fe$^{2+}$ equivalents) may be the mineral carrier of phosphite that yield phosphite during the extraction of these rocks. The chemistry of P within reduced igneous rocks (at or below the WM buffer) may include reduced P material such as CaHPO$_3$, but few speciation-specific studies have been done of P chemistry in these types of rocks.

Modern metabolic pathways implicate P$^{3+}$ as being a non-negligible portion of the total phosphorus inventory, but the absence of solid P$^{3+}$ materials as sources of this ion for life has drawn questions as to the environmental phosphite reservoir[15]. Thus a careful investigation of the anionic character of P within reduced material—though not necessarily as reduced as investigated here—might reveal other forms of crystalline reduced P minerals, many of which may have been important in the development of life on the Earth[52], or in the supply of nutrients to modern environments[18].

## Methods

**SEM**. The instrument used was a Zeiss–EVO MA15 Scanning Electron Microscope coupled with an Oxford INCA250 energy-dispersive spectrometer, operating at 25 kV accelerating potential, 500 pA probe current, 2500 cps as average count rate on the whole spectrum, and a counting time of 500 s. Samples were sputter-coated with 30-nm-thick carbon film. X-ray maps were collected with acquisition times of 10 ms per pixel.

**X-ray diffraction**. Studies were performed at the CRIST, Centro di Studi per la Cristallografia Strutturale, Department of Chemistry, Università di Firenze, Italy. A small fragment (size about 14 × 13 × 11 μm) was extracted from the polished section (Supplementary Fig. 2) under a reflected light microscope and mounted on a 5-μm-diameter carbon fiber, which was, in turn, attached to a glass rod. Single-crystal X-ray diffraction data were collected by means of a Bruker D8 Venture equipped with a Photon II CCD detector, using graphite-monochromatized MoKα radiation ($\lambda = 0.71073$ Å), exposure time of 30 s per frame and a detector-to-sample distance of 7 cm. Intensity data were integrated and corrected for Lorentz-polarization and absorption with the Bruker software packages. A full Ewald sphere was collected up to $2\theta = 70°$ at room temperature.

**LA-ICPMS**. Laser Ablation Inductively Coupled Plasma Mass Spectrometry was performed using a New Wave Research (esi) laser (NWR-213) attached to a Perkin Elmer Nexion 2000P Quadrupole ICPMS. Two samples of groundmass glass were analyzed by LA-ICPMS. Glass samples were chosen as fragments of the NPR fulgurite, then sequentially ground and polished to a 0.05-μm alumina grit and were mounted using putty. Four points on each glass were ablated and measured for the abundance of $^{29}$Si, $^{48}$Ti, $^{27}$Al, $^{23}$Na, $^{57}$Fe, $^{55}$Mn, $^{43}$Ca, $^{39}$K, $^{31}$P, and $^{11}$B (each chosen to minimize the effect of other isobars), which were referenced to USGS glasses BCR-2G, BHVO-2G, and BIR-1G[53] mounted in epoxy to constrain major element abundances that were normalized to 100% by weight.

**Thermodynamic calculations**. Data for the Gibbs free energy of various species that determine oxygen fugacity as buffers (Fe$_2$O$_3$, Fe$_3$O$_4$, FeO, Fe, FeSi, FeSi$_2$, CaSiO$_3$, Ca$_3$(PO$_4$)$_2$, Fe$_3$P, SiO$_2$, C$_6$H$_{12}$O$_6$, CO$_2$(g), and O$_2$(g)) were retrieved from HSC Chemistry (Outokompu Research Oy, v. 7.1, see details in refs. [54,55]). Data for the CaHPO$_3$/CaHPO$_4$ are from Pasek and Block[19] and were calculated by determining the difference in solubility between CaHPO$_3$ and CaHPO$_4$ (monetite), then assuming the C$_P$s of both were similar, allowing extension of the buffer to high temperature. While such a method is likely to be less accurate at high temperature, it provides an approximation of the $f_{O_2}$ buffer that may be accurate enough on the logarithmic scale to make conclusions about the relative importance of various reactions.

**CaHPO₄ reduction experiments**. CaHPO₄ (99% from Fisher Scientific, 0.031 g) was mixed with Fe and Si powder (0.1 g each) to give a 0.1:1:2 mixture of CaHPO₄:Fe:Si by mole. This mixture was heated in an alumina crucible under N₂ gas at 1000 °C for 8 h. The resulting powder was extracted with a solution of Na₄EDTA in water (10 mL) for 1 week, and analyzed by $^{31}$P NMR (details in[49]) for 10,000 scans in $^{1}$H-coupled mode.

**Nanotomography**. High-resolution multiscale nanotomography was carried out by means of a Bruker Skyscan 2214. The sample was fixed on the sample holder with dental wax, and the entire volume was scanned in one field of view to visualize pores and different density phases inside the sample. The Paganin Phase Retrieval algorithm was applied to the projection images to enhance the different density feature inside the sample.

## Data availability
Data generated or analyzed during this study are included in this published article (and its supplementary information files; https://www.researchgate.net/project/New-Port-Richey-Fulgurite). Raw data for charts (NMR fid, Laser Ablation ICPMS data, oxygen fugacity vs. temperature data for buffers, and the thermal maximum temperature profile) may be found publically at https://www.researchgate.net/project/New-Port-Richey-Fulgurite.

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

## Acknowledgements

The authors thank Z. Atlas and J. Henson for assistance with LA-ICPMS analysis. The authors acknowledge funding from NASA Exobiology grant 80NSSC23K0019 (M.A.P. and T.F.) and MIUR-PRIN2017 project "TEOREM —deciphering geological processes using Terrestrial and Extraterrestrial ORE Minerals", prot. 2017AK8C32 (LB). No permissions were required for sampling the New Port Richey fulgurite.

## Author contributions

Conceptualization: M.A.P.; L.B.: methodology: M.A.P., L.B., and T.F.; investigation: M.A.P., L.B., and T.F.; visualization: M.A.P. and L.B.; writing: M.A.P. and L.B.

## Competing interests

The authors declare no competing interests.
