## [Peer Review File · Communications Earth & Environment]

20th Oct 22

Dear Professor Pasek,

Your manuscript titled "Routes to reduction of phosphate by high-energy events" has now been seen by 3 reviewers, whose comments are appended below. You will see that they find your work of some potential interest. However, they have raised quite substantial concerns that must be addressed. In light of these comments, we cannot accept the manuscript for publication at least in its current form.

We hope you will find the reviewers' comments useful as you decide how to proceed. Should you decide to address these criticisms, please ensure that the following editorial thresholds are met:

- Provide further evidence that the results presented in the manuscript are sufficiently significant compared to previous studies and have substantial implications for our understanding of the global P cycle, as mentioned by the second and the last reviewers.
- Provide additional information about the methodology used in the study so that the results can be repeated and reproduced, as requested by the first reviewer.

If the revision process takes significantly longer than three months, we will be happy to reconsider your paper at a later date, as long as nothing similar has been accepted for publication at Communications Earth & Environment or published elsewhere in the meantime.

We understand that due to the current global situation, the time required for revision may be longer than usual. We would appreciate it if you could keep us informed about an estimated timescale for resubmission, to facilitate our planning. Of course, if you are unable to estimate, we are happy to accommodate necessary extensions nevertheless.

Please use the following link to submit your revised manuscript, point-by-point response to the reviewers' comments with a list of your changes to the manuscript text (which should be in a separate document to any cover letter) and any completed checklist:

[link redacted]

Please do not hesitate to contact me if you have any questions or would like to discuss the required revisions further. Thank you for the opportunity to review your work.

Best regards,

-Moji

Mojtaba Fakhraee, PhD
Editorial Board Member
Communications Earth & Environment
orcid.org/0000-0002-2461-6374

Clare Davis, PhD
Senior Editor
Communications Earth & Environment

EDITORIAL POLICIES AND FORMAT

If you decide to resubmit your paper, please ensure that your manuscript complies with our editorial policies and complete and upload the checklist below as a Related Manuscript file type with the revised article:

Editorial Policy Policy requirements (Download the link to your computer as a PDF.)

For your information, you can find some guidance regarding format requirements summarized on the following checklist:(<https://www.nature.com/documents/commsj-phys-style-formatting-checklist-article.pdf>) and formatting guide (<https://www.nature.com/documents/commsj-phys-style-formatting-guide-accept.pdf>).

REVIEWER COMMENTS:

Reviewer #1 (Remarks to the Author):

In this paper, the authors have characterized a fulgurite from New Port Richey, Florida, USA. I found the manuscript interesting and well detailed in the experimental description. Overall, the quality of the study is very good and innovative. The formation of P-rich metal phosphides is intriguing as the phenomenon is not well known. The introduction is informative. The results obtained from each method are described clearly. The discussion of results provides a good viewpoint on the new discoveries compared to the literature but needs some improvements. The generic terms "high temperature", "small size", "short duration" used in several parts of the discussion, decrease the

applicability and repeatability of the study, particularly when referred to the laboratory simulation of phosphite formation.

For these reasons, I recommend the publication with major revision. Please consider the comments below to revise your manuscript:

Line 12. Change "mineralogy of" with "in minerals"

Line 60. The sentence seems not clear. I suggest changing "caused by combustion of in situ organics, or through galvanic reduction" with "generated by a combustion reaction of in situ organics."

Line 62. What do you mean with "corrode"? How about "which react reducing to phosphite"

Line 80. What types of lithology? Or do you mean other minerals? Please be specific.

Line 88. Change "X-ray" with "X-ray diffraction"

Line 172. How much rapid? What is the time frame of the synthetic phase produced experimentally?

Line 178. How much "high temperature"? Please be specific.

Line 179. "The spherical shape also indicates that iron melted ($T > 1538$ °C)". Is this statement supported by data? I do not see a clear relationship between the shape and the melting point. Could you please clarify?

Line 187. Higher temperatures is a generic term. What temperatures specifically?

Line 194. Change "proceed to completion" with "complete"

Line 194. "relatively short duration". How much short? Please be specific.

Reviewer #2 (Remarks to the Author):

Routes to reduction of phosphate by high-energy events

Summary

Bindi et al document the discovery of a novel phosphite-bearing mineral in a fulgurite. This is the first mineral documented with a P oxidation state intermediate between phosphide and phosphate. The authors note the importance of this discovery, as lightning strikes may be an important route for reduced P supply to the environment (both now and in the past).

I am not an expert in fulgurite formation or high-temperature phase equilibria, so I cannot comment in great detail on those technical aspects of the study. I did find the writing clear and well-organized throughout, such that the aims of approach of the study were quite easy to follow. The documentation of the fulgurite geochemistry appears to be sound, and the authors are well-positioned to recognize the importance of this finding given a history of prior work on the topic.

I have only minor general and specific comments below, which I hope the authors will consider to make the paper yet clearer and more complete. I think with a bit more discussion of the implications of this discovery, it would indeed be of interest to the wider community.

General comments

What is/are the reductant(s) that reduce P and Fe (+ other elements) during a lightning

strike? Given that this is occurring in an O₂-rich atmosphere, it would seem likely that high-energy events should oxidize elements. So I take it the reductant is likely from a geological source (e.g., organic matter)? It would be helpful to specify this for the reader to grasp the redox balance of the fulgurite system. [note: I see at one point the brushite reduction eqn shows H₂O as the reductant- if this is true for all P and Fe reduction reactions of interest here, it would be helpful to mention this earlier]

At the end of the text, it might be helpful to provide a few more sentences about how this finding (in both character and magnitude) impacts our understanding of reduced P supply to the environment. The authors close by acknowledging this implication, but it would be of interest to the reader to know, e.g., 1) if this type of environment is relatively common, or this style of fulgurite formation is rather rare, 2) how this flux pathway quantitatively scales with other possible sources of reduced P (or phosphite specifically) to Earth's surface, 3) the role this might have played (or not) in supplying phosphite to early Earth environments.

Line-by-line comments

Line 129: "source of the discharge to be an artificial source"

Lines 124-139: Regarding the timescale argument for a lightning strike versus downed powerline- is it possible that a downed powerline could provide a short-lived (<100 sec) heating timescale as well? For instance, if it only briefly touched the surface like a lightning strike?

Lines 227-233: What was the process to prepare the fulgurite hand sample for laser ablation (polishing, mounting, etc)?

Reviewer #3 (Remarks to the Author):

This study provides a unique glimpse into a relatively rare process--how lightning strikes might modify phosphorus chemistry and mineralogy of substrates. The arguments laid out here, based both on a natural sample as well as a synthetic high temperature experimental example, indicate that reduced forms of phosphorus can indeed be formed in unique environments. The authors argue that this result has implications for past dynamics of phosphorus cycling on Earth as well as several astrobiological examples.

I find that although the analytical approaches applied, including microscopy, geochemistry, mineralogy, and synthetic analog experiments, are robust and definitive, the overall results document a relatively rare occurrence that is not a dominant feature of the phosphorus cycle and still has many uncertainties wrt kinetics of the high temperature triggering event. For example, the reaction front and triggering temperature of the purported lightning strike that formed the fulgurite examined here are very poorly defined and not captured adequately in the summary model presented in Figure 5. The temperature range and thermal front of the lightning strike seems poorly described, leaving only illusive indicators of processes that occur in this transition zone (including the metallic phosphide droplets). It is all very scientifically interesting, to be sure, but mostly to a niche audience of geochemists and mineralogists.

RESPONSE TO REVIEWERS

Reviewer #1 (Remarks to the Author):

In this paper, the authors have characterized a fulgurite from New Port Richey, Florida, USA. I found the manuscript interesting and well detailed in the experimental description.

Overall, the quality of the study is very good and innovative. The formation of P-rich metal phosphides is intriguing as the phenomenon is not well known.

The introduction is informative. The results obtained from each method are described clearly. The discussion of results provides a good viewpoint on the new discoveries compared to the literature but needs some improvements. The generic terms "high temperature", "small size", "short duration" used in several parts of the discussion, decrease the applicability and repeatability of the study, particularly when referred to the laboratory simulation of phosphite formation.

>>We have attempted to address these generic terms in the current manuscript, better framing known and unknown parameters. To some extent this ambiguity deals with the ranges of temperatures that are experienced during fulgurite formation, which often vary by 1000°C or more. Namely, in our summary figure (7), we've added known temperature points.

For these reasons, I recommend the publication with major revision. Please consider the comments below to revise your manuscript:

Line 12. Change "mineralogy of" with "in minerals"

>>Changed

Line 60. The sentence seems not clear. I suggest changing "caused by combustion of in situ organics, or through galvanic reduction" with "generated by a combustion reaction of in situ organics."

>>Changed

Line 62. What do you mean with "corrode"? How about "which react reducing to phosphite"

>> modified to which react with water to release phosphite as a metastable solute". Notably this process is a "corrosion" in that there's a transfer of electrons out of the metal and a formation of an oxide/oxyhydroxide on the surface of the phosphide, but this still includes the description that the phosphide reaction with water results in phosphite and not phosphate.

Line 80. What types of lithology? Or do you mean other minerals? Please be specific.

>>Changed to "The New Port Richey fulgurite bears both silicate and reduced (silicide) compositional regions."

Line 88. Change "X-ray" with "X-ray diffraction"

>>Done

Line 172. How much rapid? What is the time frame of the synthetic phase produced experimentally?

>> added "after 8 hours" and "on the timescale of 60 seconds or less" to these sentences.

Line 178. How much "high temperature"? Please be specific.

>> added (T>1538°C)

Line 179. "The spherical shape also indicates that iron melted (T>1538 °C)". Is this statement supported by data? I do not see a clear relationship between the shape and the melting point. Could you please clarify?

>> The sphericity of the metals indicates that they were at least molten. We've placed in a figure reference to clarify which spherules we are discussing.

Line 187. Higher temperatures is a generic term. What temperatures specifically?

>> Added ">200°C" and a reference

Line 194. Change "proceed to completion" with "complete"

>> done

Line 194. "relatively short duration". How much short? Please be specific.

>> added "<1 minute"

Reviewer #2 (Remarks to the Author):

Routes to reduction of phosphate by high-energy events

Summary

Bindi et al document the discovery of a novel phosphite-bearing mineral in a fulgurite. This is the first mineral documented with a P oxidation state intermediate between phosphide and phosphate. The authors note the importance of this discovery, as lightning strikes may be an important route for reduced P supply to the environment (both now and in the past).

I am not an expert in fulgurite formation or high-temperature phase equilibria, so I cannot comment in great detail on those technical aspects of the study. I did find the writing clear and well-organized throughout, such that the aims of approach of the study were quite easy to follow. The documentation of the fulgurite geochemistry appears to be sound, and the authors are well-positioned to recognize the importance of this finding given a history of prior work on the topic.

I have only minor general and specific comments below, which I hope the authors will consider to make the paper yet clearer and more complete. I think with a bit more discussion of the implications of this discovery, it would indeed be of interest to the wider community.

General comments

What is/are the reductant(s) that reduce P and Fe (+ other elements) during a lightning strike? Given that this is occurring in an O₂-rich atmosphere, it would seem likely that high-energy events should oxidize elements. So I take it the reductant is likely from a geological source (e.g., organic matter)? It would be helpful to specify this for the reader to grasp the redox balance of the fulgurite system. [note: I see at one point the brushite reduction eqn shows H₂O as the reductant- if this is true for all P and Fe reduction reactions of interest here, it would be helpful to mention this earlier]

>> We do believe the reductant is organic matter, likely organics within the soil that combusted during the electric discharge. Specifically, we believe it's a tree root. We've approximated this compositionally as glucose (C₆H₁₂O₆). We've added some thermodynamics to the SI to show this is feasible (as long as CO₂ and water vapor escape), as the equilibrium constant values are sufficiently large that the reduction of Fe- and Si-oxides to FeSi is feasible.

At the end of the text, it might be helpful to provide a few more sentences about how this finding (in both character and magnitude) impacts our understanding of reduced P supply to the environment. The authors close by acknowledging this implication, but it would be of interest to the reader to know, e.g., 1) if this type of environment is relatively common, or this style of fulgurite formation is rather rare, 2) how this flux pathway quantitatively scales with other possible sources of reduced P (or phosphite specifically) to Earth's surface, 3) the role it might have played (or not) in supplying phosphite to early Earth environments.

>>The general formation of iron silicides in fulgurites is somewhat uncommon (most are just silica-rich glass), and hence this specific occurrence is likely not widespread. However, other fulgurites have had phosphite as a component of them (without silicides) but the mineral carrier could not be identified. We've added text to the discussion that also provides Hess et al. (2021)'s argument that P from lightning could have been a major source of P on the early earth, and Pasek and Block (2009)'s constraints of this process on the modern biosphere (relatively minor). That said, the findings presented here show that reduction of Ca-P5+ to Ca-P3+ occurs readily (and quickly, over <1 minute) and may suggest that there are many other instances of phosphite in the environment. We've added more discussion to the end with these comments.

Line-by-line comments

Line 129: "source of the discharge to be an artificial source"

>> Done.

Lines 124-139: Regarding the timescale argument for a lightning strike versus downed powerline- is it possible that a downed powerline could provide a short-lived (<100 sec) heating timescale as well? For instance, if it only briefly touched the surface like a lightning strike?

>> A brief contact? This is a good question, but we think no. Most artificial discharge sources would probably have had to have taken place over tens of minutes to hours of direct contact in order to create the fulgurite of the size we see here (500 g). This is especially true since the NPR fulgurite was found in a residential neighborhood (where the voltage carried by the power lines is lower than industrial or long-distance requirements). However, we cannot fully exclude this as an option and are extrapolating experiments at 1000°C to 1240°C so we've attempted to be cautious in our description of the fulgurite's origin though we lean towards natural.

Lines 227-233: What was the process to prepare the fulgurite hand sample for laser ablation (polishing, mounting, etc)?

>>Added (polished sequentially to 50 nm alumina grit, etc.)

Reviewer #3 (Remarks to the Author):

This study provides a unique glimpse into a relatively rare process--how lightning strikes might modify phosphorus chemistry and mineralogy of substrates. The arguments laid out here, based both on a natural sample as well as a synthetic high temperature experimental example, indicate that reduced forms of phosphorus can indeed be formed in unique environments. The authors argue that this result has implications for past dynamics of phosphorus cycling on Earth as well as several astrobiological examples.

I find that although the analytical approaches applied, including microscopy, geochemistry, mineralogy, and synthetic analog experiments, are robust and definitive, the overall results document a relatively rare occurrence that is not a dominant feature of the phosphorus cycle and still has many uncertainties wrt kinetics of the high temperature triggering event. For example, the reaction front and triggering temperature of the purported lightning strike that formed the fulgurite examined here are very poorly defined and not captured adequately in the summary model presented in Figure 5. The temperature range and thermal front of the lightning strike seems poorly described, leaving only illusive indicators of processes that occur in this transition zone (including the metallic phosphide droplets). It is all very scientifically interesting, to be sure, but mostly to a niche audience of geochemists and mineralogists.

>> Reviewer 1 requested clearer descriptions of temperatures reached and we have attempted to fill these in (see the new caption for Figure 5 (now 7)). We've also added some more reaction thermodynamics as requested by reviewer 2.

We have added text as well to justify the relevance of finding a P³⁺ mineral to the phosphorus biogeochemical cycle. We do agree that this specific occurrence is rather niche: extreme reduction to make iron silicides is not a common process. However, the identification of a solid, naturally-occurring phosphite material is impactful, given the importance of P to biology, and the fact that many forms of life can use P³⁺ as sole P sources (despite having no previously reported solid reservoirs of this material). This is not just the potential discovery of a new mineral (which wouldn't be terribly impactful or exciting) but a new mineral group, and of an element that has biological importance. The identification of this crystal form of P³⁺ may allow for future P mineral investigations of other samples that might yield identical occurrences in more widespread rocks, or of minerals that are similar (e.g., FeHPO₃, MgHPO₃). For this reason we feel the finding extends beyond a niche "weird fulgurite mineralogy".

24th Jan 23

Dear Professor Pasek,

Please allow us to apologise for the delay in sending a decision on your manuscript titled "Routes to reduction of phosphate by high-energy events". It has now been seen by our reviewers, whose comments appear below. In light of their advice I am delighted to say that we are happy, in principle, to publish a suitably revised version in Communications Earth & Environment under the open access CC BY license (Creative Commons Attribution v4.0 International License).

We therefore invite you to revise your paper one last time to address the remaining concerns of our reviewers and to soften your claims on the scale of importance for our conception of the global phosphorus cycle. At the same time we ask that you edit your manuscript to comply with our format requirements and to maximise the accessibility and therefore the impact of your work.

EDITORIAL REQUESTS:

*****Please take care to match our formatting and policy requirements. We will check revised manuscript and return manuscripts that do not comply. Such requests will lead to delays. *****

SUBMISSION INFORMATION:

OPEN ACCESS:

Communications Earth & Environment is a fully open access journal. Articles are made freely accessible on publication under a [CC BY license](http://creativecommons.org/licenses/by/4.0) (Creative Commons Attribution 4.0 International License). This license allows maximum dissemination and re-use of open access materials and is preferred by many research funding bodies.

For further information about article processing charges, open access funding, and advice and support from Nature Research, please visit <https://www.nature.com/commsenv/article-processing-charges>

At acceptance, you will be provided with instructions for completing this CC BY license on behalf of all authors. This grants us the necessary permissions to publish your paper. Additionally, you will be asked to declare that all required third party permissions have been obtained, and to provide billing information in order to pay the article-processing charge (APC).

[link redacted]

Best regards,

Joe Aslin
Senior Editor,
Communications Earth & Environment
<https://www.nature.com/commsenv/>
Twitter: @CommsEarth

REVIEWERS' COMMENTS:

Reviewer #1 (Remarks to the Author):

I revised the manuscript COMMSENV-22-0890A entitled "Routes to reduction of phosphate by high-energy events". The authors took my proposed changes into account when producing the revised version. However, there are still some changes that I would suggest, please see below. As for these reasons, I suggest the publication of the paper with minor revisions.

Line 21-23. the sentence is not clear. Also, what do you mean by much higher redox state? Please be specific.

Line 23-24. This sentence is vague. How does this assumption be proved?

Line 53. "...exist in solid form of some sort" is generic. Please be specific. Do you have a reference?

Line 53-55. this sentence seems to contradict the previous one where it said that "reduced P oxyacids must exist in solid form of some sort". Please rephrase.

Line 223 and 237. Please change "earth" with "Earth"

Reviewer #2 (Remarks to the Author):

The authors have adequately addressed my comments on the previous iteration of the manuscript. I am happy to see the paper published pending the approval of the other reviewers.

Reviewer #3 (Remarks to the Author):

I have reviewed the revised manuscript, and the authors have addressed a number of reviewer comments, including one of my own focused on restructuring figure 5 from original manuscript. My overall impressions remain unchanged--this is a well-researched and sound contribution which focuses on a relatively rare occurrence. This does not reshape our conception of the phosphorus cycle on Earth, but it does provide insight into the processes yielding a new and unique class of P-bearing minerals.

We thank the reviewers for their critique and comments. We've made the following changes to the manuscript in light of these recommendations.

Reviewer #1 (Remarks to the Author):

I revised the manuscript COMMSENV-22-0890A entitled "Routes to reduction of phosphate by high-energy events". The authors took my proposed changes into account when producing the revised version. However, there are still some changes that I would suggest, please see below. As for these reasons, I suggest the publication of the paper with minor revisions.

>>Thank you!

Line 21-23. the sentence is not clear. Also, what do you mean by much higher redox state? Please be specific.

>>Because we have had to shrink the abstract to 150 words, this text was removed.

Line 23-24. This sentence is vague. How does this assumption be proved?

>>Because we have had to shrink the abstract to 150 words, this text was removed.

Line 53. "...exist in solid form of some sort" is generic. Please be specific. Do you have a reference?

>>rewritten as "Therefore, reduced P oxyacids must exist in solid form of some sort in the environment, either as a pure compound such as CaHPO_3 , or with the phosphite exchanging for other ions such as phosphate or sulfate."

Right now there are no references for either, which is part of the goal of the present paper: to show a solid phosphite phase that is plausible in nature.

Line 53-55. this sentence seems to contradict the previous one where it said that "reduced P oxyacids must exist in solid form of some sort". Please rephrase.

>>Rephrased as "The failure to identify a specific mineral form of reduced phosphorus oxyacid in the environment is problematic given the widespread evidence for phosphorus redox in biogeochemistry (16), but may be due instead to an incomplete search for such material."

In this case, there's a contradiction—phosphite is extracted from rocks, but isn't known in mineral form. The prior point is there to suggest that that's because it's not been looked for before.

Line 223 and 237. Please change "earth" with "Earth"

>>Done

Reviewer #2 (Remarks to the Author):

The authors have adequately addressed my comments on the previous iteration of the manuscript. I am happy to see the paper published pending the approval of the other reviewers.

>>Thank you!

Reviewer #3 (Remarks to the Author):

I have reviewed the revised manuscript, and the authors have addressed a number of reviewer comments, including one of my own focused on restructuring figure 5 from original manuscript. My overall impressions remain unchanged--this is a well-researched and sound contribution which focuses on a relatively rare occurrence. This does not reshape our conception of the phosphorus cycle on Earth, but it does provide insight into the processes yielding a new and unique class of P-bearing minerals.

>>Thank you- we do hope that this discovery may build into future discoveries in P mineralogy, as the present discovery is indeed quite limited.